# Optimization Methods for Sparse Pseudo-Likelihood Graphical Model Selection

**Sang-Yun Oh**
Computational Research Division
Lawrence Berkeley National Lab
syoh@lbl.gov

**Onkar Dalal**
Stanford University
onkar@alumni.stanford.edu

**Kshitij Khare**
Department of Statistics
University of Florida
kdkhare@stat.ufl.edu

**Bala Rajaratnam**
Department of Statistics
Stanford University
brajarat@stanford.edu

## Abstract

Sparse high dimensional graphical model selection is a popular topic in contemporary machine learning. To this end, various useful approaches have been proposed in the context of $\ell_1$-penalized estimation in the Gaussian framework. Though many of these inverse covariance estimation approaches are demonstrably scalable and have leveraged recent advances in convex optimization, they still depend on the Gaussian functional form. To address this gap, a convex pseudo-likelihood based partial correlation graph estimation method (CONCORD) has been recently proposed. This method uses coordinate-wise minimization of a regression based pseudo-likelihood, and has been shown to have robust model selection properties in comparison with the Gaussian approach. In direct contrast to the parallel work in the Gaussian setting however, this new convex pseudo-likelihood framework has not leveraged the extensive array of methods that have been proposed in the machine learning literature for convex optimization. In this paper, we address this crucial gap by proposing two proximal gradient methods (CONCORD-ISTA and CONCORD-FISTA) for performing $\ell_1$-regularized inverse covariance matrix estimation in the pseudo-likelihood framework. We present timing comparisons with coordinate-wise minimization and demonstrate that our approach yields tremendous payoffs for $\ell_1$-penalized partial correlation graph estimation outside the Gaussian setting, thus yielding the fastest and most scalable approach for such problems. We undertake a theoretical analysis of our approach and rigorously demonstrate convergence, and also derive rates thereof.

## 1 Introduction

Sparse inverse covariance estimation has received tremendous attention in the machine learning, statistics and optimization communities. These sparse models, popularly known as graphical models, have widespread use in various applications, especially in high dimensional settings. The most popular inverse covariance estimation framework is arguably the $\ell_1$-penalized Gaussian likelihood optimization framework as given by

$$\underset{\Omega \in \mathbf{S}^p_{++}}{\text{minimize}} \quad -\log\det\Omega + \text{tr}(S\Omega) + \lambda\|\Omega\|_1$$

where $\mathbf{S}^p_{++}$ denotes the space of $p$-dimensional positive definite matrices, and $\ell_1$-penalty is imposed on the elements of $\Omega = (\omega_{ij})_{1 \le i \le j \le p}$ by the term $\|\Omega\|_1 = \sum_{i,j}|\omega_{ij}|$ along with the scaling factor

$\lambda > 0$. The matrix $S$ denotes the sample covariance matrix of the data $\mathbf{Y} \in I\!\!R^{n \times p}$. As the $\ell_1$-penalized log likelihood is convex, the problem becomes more tractable and has benefited from advances in convex optimization. Recent efforts in the literature on Gaussian graphical models therefore have focused on developing principled methods which are increasingly more and more scalable. The literature on this topic is simply enormous and for the sake of brevity, space constraints and the topic of this paper, we avoid an extensive literature review by referring to the references in the seminal work of [1] and the very recent work of [2]. These two papers contain references to recent work, including past NIPS conference proceedings.

## 1.1 The CONCORD method

Despite their tremendous contributions, one shortcoming of the traditional approaches to $\ell_1$-penalized likelihood maximization is the restriction to the Gaussian assumption. To address this gap, a number of $\ell_1$-penalized pseudo-likelihood approaches have been proposed: SPACE [3] and SPLICE [4], SYMLASSO [5]. These approaches are either not convex, and/or convergence of corresponding maximization algorithms are not established. In this sense, non-Gaussian partial correlation graph estimation methods have lagged severely behind, despite the tremendous need to move beyond the Gaussian framework for obvious practical reasons. In very recent work, a convex pseudo-likelihood approach with good model selection properties called CONCORD [6] was proposed. The CONCORD algorithm minimizes

$$Q_{\mathrm{con}}(\Omega) = -\sum_{i=1}^{p} n \log \omega_{ii} + \frac{1}{2} \sum_{i=1}^{p} \|\omega_{ii} \mathbf{Y}_i + \sum_{j \neq i} \omega_{ij} \mathbf{Y}_j\|_2^2 + n\lambda \sum_{1 \leq i < j \leq p} |\omega_{ij}| \qquad (1)$$

via cyclic coordinate-wise descent that alternates between updating off-diagonal elements and diagonal elements. It is straightforward to show that operators $T_{ij}$ for updating $(\omega_{ij})_{1 \leq i < j \leq p}$ (holding $(\omega_{ii})_{1 \leq i \leq p}$ constant) and $T_{ii}$ for updating $(\omega_{ii})_{1 \leq i \leq p}$ (holding $(\omega_{ij})_{1 \leq i < j \leq p}$ constant) are given by

$$(T_{ij}(\Omega))_{ij} = \frac{S_\lambda \left( - \left( \sum_{j' \neq j} \omega_{ij'} s_{jj'} + \sum_{i' \neq i} \omega_{i'j} s_{ii'} \right) \right)}{s_{ii} + s_{jj}} \qquad (2)$$

$$(T_{ii}(\Omega))_{ii} = \frac{- \sum_{j \neq i} \omega_{ij} s_{ij} + \sqrt{\left( \sum_{j \neq i} \omega_{ij} s_{ij} \right)^2 + 4 s_{ii}}}{2 s_{ii}}. \qquad (3)$$

This coordinate-wise algorithm is shown to converge to a global minima though no rate is given [6]. Note that the equivalent problem assuming a Gaussian likelihood has seen much development in the last ten years, but a parallel development for the recently introduced CONCORD framework is lacking for obvious reasons. We address this important gap by proposing state-of-the-art proximal gradient techniques to minimize $Q_{\mathrm{con}}$. A rigorous theoretical analysis of the pseudo-likelihood framework and the associated proximal gradient methods which are proposed is undertaken. We establish rates of convergence and also demonstrate that our approach can lead to massive computational speed-ups, thus yielding extremely fast and principled solvers for the sparse inverse covariance estimation problem outside the Gaussian setting.

## 2 CONCORD using proximal gradient methods

The penalized matrix version the CONCORD objective function in (1) is given by

$$Q_{\mathrm{con}}(\Omega) = \frac{n}{2} \left[ -\log |\Omega_D^2| + \mathrm{tr}(\mathbf{S}\Omega^2) + \lambda \|\Omega_X\|_1 \right]. \qquad (4)$$

where $\Omega_D$ and $\Omega_X$ denote the diagonal and off-diagonal elements of $\Omega$. We will use the notation $A = A_D + A_X$ to split any matrix $A$ into its diagonal and off-diagonal terms.

This section proposes a scalable and thorough approach to solving the CONCORD objective function using recent advances in convex optimization and derives rates of convergence for such algorithms. In particular, we use proximal gradient-based methods to achieve this goal and demonstrate the efficacy of such methods for the non-Gaussian graphical modeling problem. First, we propose CONCORD-ISTA and CONCORD-FISTA in section 2.1: methods which are inspired by the iterative soft-thresholding algorithms in [7]. We undertake a comprehensive treatment of the CONCORD

optimization problem by also investigating the dual of the CONCORD problem. Other popular methods in the literature, including the potential use of alternating minimization algorithm and the second order proximal Newtons method are considered in Supplemental section A.8.

## 2.1 Iterative Soft Thresholding Algorithms: CONCORD-ISTA, CONCORD-FISTA

The iterative soft-thresholding algorithms (ISTA) have recently gained popularity after the seminal paper by Beck and Teboulle [7]. The ISTA methods are based on the Forward-Backward Splitting method from [8] and Nesterov's accelerated gradient methods [9] using soft-thresholding as the proximal operator for the $\ell_1$-norm. The essence of the proximal gradient algorithms is to divide the objective function into a smooth part and a non-smooth part, then take a proximal step (w.r.t. the non-smooth part) in the negative gradient direction of the smooth part. Nesterov's accelerated gradient extension [9] uses a combination of gradient and momentum steps to achieve accelerated rates of convergence. In this section, we apply these methods in the context of CONCORD which also has a composite objective function.

The matrix CONCORD objective function (4) can be split into a smooth part $h_1(\Omega)$ and a non-smooth part $h_2(\Omega)$:

$$h_1(\Omega) = -\log \det \Omega_D + \frac{1}{2} \operatorname{tr}(\Omega S \Omega), \quad h_2(\Omega) = \lambda \|\Omega_X\|_1. \tag{5}$$

The gradient and hessian of the smooth function $h_1$ are given by

$$\nabla h_1(\Omega) = \Omega_D^{-1} + \frac{1}{2} \left(S\Omega^T + \Omega S\right),$$

$$\nabla^2 h_1(\Omega) = \sum_{i=1}^{i=p} \omega_{ii}^{-2} \left[e_i e_i^T \otimes e_i e_i^T\right] + \frac{1}{2} \left(S \otimes I + I \otimes S\right), \tag{6}$$

where $e_i$ is a column vector of zeros except for a one in the $i$-th position.

The proximal operator for $h_2$ is given by element-wise soft-thresholding operator $\mathcal{S}_\lambda$ as

$$\operatorname{prox}_{h_2}(\Omega) = \arg\min_{\Theta} \left\{ h_2(\Theta) + \frac{1}{2} \|\Omega - \Theta\|_F^2 \right\}$$
$$= \mathcal{S}_\Lambda(\Omega) = \operatorname{sign}(\Omega) \max\{|\Omega| - \Lambda, 0\}, \tag{7}$$

where $\Lambda$ is a matrix with 0 diagonal and $\lambda$ for each off-diagonal entry. The details of the proximal gradient algorithm CONCORD-ISTA are given in Algorithm 1, and the details of the accelerated proximal gradient algorithm CONCORD-FISTA are given in Algorithm 2.

## 2.2 Choice of step size

In the absence of a good estimate of the Lipschitz constant $L$, the step size for each iteration of CONCORD-ISTA and CONCORD-FISTA is chosen using backtracking line search. The line search for iteration $k$ starts with an initial step size $\tau_{(k,0)}$ and reduces the step with a constant factor $c$ until the new iterate satisfies the sufficient descent condition:

$$h_1(\Omega^{(k+1)}) \leq \mathcal{Q}(\Omega^{(k+1)}, \Omega^{(k)}) \tag{8}$$

where,

$$\mathcal{Q}(\Omega, \Theta) = h_1(\Theta) + \operatorname{tr}\left((\Omega - \Theta)^T \nabla h_1(\Theta)\right) + \frac{1}{2\tau} \|\Omega - \Theta\|_F^2.$$

In section 4, we have implemented algorithms choosing the initial step size in three different ways: (a) a constant starting step size (=1), (b) the feasible step size from the previous iteration $\tau_{k-1}$, (c) the step size heuristic of Barzilai-Borwein. The Barzilai-Borwein heuristic step size is given by

$$\tau_{k+1,0} = \frac{\operatorname{tr}\left((\Omega^{(k+1)} - \Omega^{(k)})^T (\Omega^{(k+1)} - \Omega^{(k)})\right)}{\operatorname{tr}\left((\Omega^{(k+1)} - \Omega^{(k)})^T (G^{(k+1)} - G^{(k)})\right)}. \tag{9}$$

This is an approximation of the secant equation which works as a proxy for second order information using successive gradients (see [10] for details).

| **Algorithm 1** CONCORD-ISTA | **Algorithm 2** CONCORD-FISTA |
|---|---|
| Input: sample covariance matrix $S$, penalty $\Lambda$ | Input: sample covariance matrix $S$, penalty $\Lambda$ |
| Set: $\Omega^{(0)} \in \mathbb{S}_+^p$, $\tau_{(0,0)} \le 1$, $c < 1$, $\Delta_{\text{subg}} = 1$ | Set: $(\Theta^{(1)} =)\Omega^{(0)} \in \mathbb{S}_+^p$, $\alpha_1 = 1$, $\tau_{(0,0)} \le 1$, |
| **while** $\Delta_{\text{subg}} > \epsilon_{\text{subg}}$ **do** | $\qquad c < 1$, $\Delta_{\text{subg}} = 1$. |
| $\quad G^{(k)} = -\left(\Omega_{\text{D}}^{(k)}\right)^{-1} + \frac{1}{2}\left(S\,\Omega^{(k)} + \Omega^{(k)}S\right)$ | **while** $\Delta_{\text{subg}} > \epsilon_{\text{subg}}$ **do** |
| $\quad$ Take largest $\tau_k \in \{c^j \tau_{(k,0)}\}_{j=0,1,\dots}$ s.t. | $\quad G^{(k)} = -\left(\Theta_{\text{D}}^{(k)}\right)^{-1} + \frac{1}{2}\left(S\Theta^{(k)} + \Theta^{(k)}S\right)$ |
| $\qquad \Omega^{(k+1)} = \mathcal{S}_{\tau_k \Lambda}\left(\Omega^{(k)} - \tau_k G^{(k)}\right) \vdash (8).$ | $\quad$ Take largest $\tau_k \in \{c^j \tau_{(k,0)}\}_{j=0,1,\dots}$ s.t. |
| $\quad$ *Compute:* $\tau_{(k+1,0)}$ | $\qquad \Omega^{(k)} = \mathcal{S}_{\tau_k \Lambda}\left(\Theta^{(k)} - \tau_k G^{(k)}\right) \vdash (8)$ |
| $\quad$ *Compute:* $\Delta_{\text{subg}}$[1] | $\quad \alpha_{k+1} = (1 + \sqrt{1 + 4\alpha_k{}^2})/2$ |
| **end while** | $\quad \Theta^{(k+1)} = \Omega^{(k)} + \left(\frac{\alpha_k - 1}{\alpha_{k+1}}\right)\left(\Omega^{(k)} - \Omega^{(k-1)}\right)$ |
| | $\quad$ *Compute:* $\tau_{(k+1,0)}$ |
| 1: $\Delta_{\text{subg}} = \dfrac{\|\nabla h_1(\Omega^{(k)}) + \partial h_2(\Omega^{(k)})\|}{\|\Omega^{(k)}\|}$ | $\quad$ *Compute:* $\Delta_{\text{subg}}$[1] |
| | **end while** |

## 2.3 Computational complexity

After the one time calculation of $S$, the most significant computation for each iteration in CONCORD-ISTA and CONCORD-FISTA algorithms is the matrix-matrix multiplication $W = S\Omega$ in the gradient term. If $s$ is the number of non-zeros in $\Omega$, then $W$ can be computed using $\mathcal{O}(sp^2)$ operations if we exploit the extreme sparsity in $\Omega$. The second matrix-matrix multiplication for the term $\text{tr}(\Omega(S\Omega))$ can be computed efficiently using $\text{tr}(\Omega W) = \sum \omega_{ij} w_{ij}$ over the set of non-zero $\omega_{ij}$'s. This computation only requires $\mathcal{O}(s)$ operations. The remaining computations are all at the element level which can be completed in $\mathcal{O}(p^2)$ operations. Therefore, the overall computational complexity for each iteration reduces to $\mathcal{O}(sp^2)$. On the other hand, the proximal gradient algorithms for the Gaussian framework require inversion of a full $p \times p$ matrix which is non-parallelizable and requires $\mathcal{O}(p^3)$ operations. The coordinate-wise method for optimizing CONCORD in [6] also requires cycling through the $p^2$ entries of $\Omega$ in specified order and thus does not allow parallelization. In contrast, CONCORD-ISTA and CONCORD-FISTA can use 'perfectly parallel' implementations to distribute the above matrix-matrix multiplications. At no step do we need to keep all of the dense matrices $S, S\Omega, \nabla h_1$ on a single machine. Therefore, CONCORD-ISTA and CONCORD-FISTA are scalable to any high dimensions restricted only by the number of machines.

## 3 Convergence Analysis

In this section, we prove convergence of CONCORD-ISTA and CONCORD-FISTA methods along with their respective convergence rates of $\mathcal{O}(1/k)$ and $\mathcal{O}(1/k^2)$. We would like to point out that, although the authors in [6] provide a proof of convergence for their coordinate-wise minimization algorithm for CONCORD, they do not provide any rates of convergence. The arguments for convergence leverage the results in [7] but require some essential ingredients. We begin with proving lower and upper bounds on the diagonal entries $\omega_{kk}$ for $\Omega$ belonging to a level set of $Q_{\text{con}}(\Omega)$. The lower bound on the diagonal entries of $\Omega$ establishes Lipschitz continuity of the gradient $\nabla h_1(\Omega)$ based on the hessian of the smooth function as stated in (6). The proof for the lower bound uses the existence of an upper bound on the diagonal entries. Hence, we prove both bounds on the diagonal entries. We begin by defining a level set $\mathcal{C}_0$ of the objective function starting with an arbitrary initial point $\Omega^{(0)}$ with a finite function value as

$$\mathcal{C}_0 = \left\{ \Omega \mid Q_{\text{con}}(\Omega) \le Q_{\text{con}}(\Omega^{(0)}) = M \right\}. \tag{10}$$

For the positive semidefinite matrix $S$, let $U$ denote $\frac{1}{\sqrt{2}}$ times the upper triangular matrix from the LU decomposition of $S$, such that $S = 2U^T U$ (the factor 2 simplifies further arithmetic). Assuming

the diagonal entries of $S$ to be strictly nonzero (if $s_{kk} = 0$, then the $k^{th}$ component can be ignored upfront since it has zero variance and is equal to a constant for every data point), we have at least one $k$ such that $u_{ki} \neq 0$ for every $i$. Using this, we prove the following theorem.

**Theorem 3.1.** *For any symmetric matrix $\Omega$ satisfying $\Omega \in \mathcal{C}_0$, the diagonal elements of $\Omega$ are bounded above and below by constants which depend only on $M$, $\lambda$ and $S$. In other words,*

$$0 < a \leq |\omega_{kk}| \leq b, \ \ \forall \ k = 1, 2, \ldots, p,$$

*for some constants $a$ and $b$. (((removed subscripts for $a$ and $b$)))*

*Proof.* (a) Upper bound: Suppose $|\omega_{ii}| = \max\{|\omega_{kk}|, \text{for } k = 1, 2, \ldots, p\}$. Then, we have

$$M = Q_{\text{con}}(\Omega^{(0)}) \geq Q_{\text{con}}(\Omega) = h_1(\Omega) + h_2(\Omega)$$
$$\geq -\log \det \Omega_D + \text{tr} \left( (U\Omega)^T (U\Omega) \right) + \lambda \|\Omega_X\|_1$$
$$= -\log \det \Omega_D + \|U\Omega\|_F^2 + \lambda \|\Omega_X\|_1. \tag{11}$$

Considering $ki^{th}$ entry in the Frobenious norm and the $i^{th}$ column in the third term we get

$$M \geq -p \log |\omega_{ii}| + \left( \sum_{j=k}^{j=p} u_{kj}\omega_{ji} \right)^2 + \lambda \sum_{j=k, j \neq i}^{j=p} |\omega_{ji}|. \tag{12}$$

Now, suppose $|u_{ki}\omega_{ii}| = z$ and $\sum_{j=k, j \neq i}^{j=p} u_{kj}\omega_{ji} = x$. Then

$$|x| \leq \sum_{j=k, j \neq i}^{j=p} |u_{kj}||\omega_{ji}| \leq \bar{u} \sum_{j=k, j \neq i}^{j=p} |\omega_{ji}|,$$

where $\bar{u} = \max\{|u_{kj}|\}$, for $j = k, \ldots, p$, $j \neq i$. Substituting in (12), for $\bar{\lambda} = \frac{\lambda}{2\bar{u}}$, we have

$$\bar{M} = M + \bar{\lambda}^2 - p \log |u_{ki}| \geq -p \log z + (z + x)^2 + 2\bar{\lambda}|x| + \bar{\lambda}^2 \tag{13}$$
$$= -p \log z + (z + x + \bar{\lambda} \text{sign}(x))^2 - 2\bar{\lambda} z \, \text{sign}(x) \tag{14}$$

Here, if $x \geq 0$, then $\bar{M} \geq -p \log z + z^2$ using the first inequality (13), and if $x < 0$, then $\bar{M} \geq -p \log z + 2\bar{\lambda}z$ using the second inequality (14). In either cases, the functions $-p \log z + z^2$ and $-p \log z + 2\bar{\lambda}z$ are unbounded as $z \to \infty$. Hence, the upper bound of $\bar{M}$ on these functions guarantee an upper bound $b$ such that $|\omega_{ii}| \leq b$. Therefore, $|\omega_{kk}| \leq b$ for all $k = 1, 2, \ldots, p$.

(b) Lower bound: By positivity of the trace term and the $\ell_1$ term (for off-diagonals), we have

$$M \geq -\log \det \Omega_D = \sum_{i=1}^{i=p} -\log |\omega_{ii}|. \tag{15}$$

The negative log function $g(z) = -\log(z)$ is a convex function with a lower bound at $z^* = b$ with $g(z^*) = -\log b$. Therefore, for any $k = 1, 2, \ldots, p$, we have

$$M \geq \sum_{i=1}^{i=p} -\log |\omega_{ii}| \geq -(p-1) \log b - \log |\omega_{kk}|. \tag{16}$$

Simplifying the above equation, we get

$$\log |\omega_{kk}| \geq -M - (p-1) \log b.$$

Therefore, $|\omega_{kk}| \geq a = e^{-M-(p-1)\log b} > 0$ serves as a lower bound for all $k = 1, 2, \ldots, p$. $\square$

Given that the function values are non-increasing along the iterates of Algorithms 1, 2 and 3, the sequence of $\Omega^{(k)}$ satisfy $\Omega^{(k)} \in \mathcal{C}_0$ for $k = 1, 2, \ldots.$. The lower bounds on the diagonal elements of $\Omega^{(k)}$ provides the Lipschitz continuity using

$$\nabla^2 h_1(\Omega^{(k)}) \preceq (a^{-2} + \|S\|_2) (I \otimes I). \tag{17}$$

Therefore, using the mean-value theorem, the gradient $\nabla h_1$ satisfies

$$\|\nabla h_1(\Omega) - \nabla h_1(\Theta)\|_F \leq L \|\Omega - \Theta\|_F, \tag{18}$$

with the Lipschitz continuity constant $L = a^{-2} + \|S\|_2$. The remaining argument for convergence follows from the theorems in [7].

**Theorem 3.2.** *([7, Theorem 3.1]). Let $\{\Omega^{(k)}\}$ be the sequence generated by either Algorithm 1 with constant step size or with backtracking line-search. Then, for the solution $\Omega^*$, for any $k \geq 1$,*

$$Q_{con}(\Omega^{(k)}) - Q_{con}(\Omega^*) \leq \frac{\alpha L \|\Omega^{(0)} - \Omega^*\|_F^2}{2k}, \tag{19}$$

*where $\alpha = 1$ for the constant step size setting and $\alpha = c$ for the backtracking step size setting.*

**Theorem 3.3.** *([7, Theorem 4.4]). Let $\{\Omega^{(k)}\}, \{\Theta^{(k)}\}$ be the sequences generated by Algorithm 2 with either constant step size or backtracking line-search. Then, for the solution $\Omega^*$, for any $k \geq 1$,*

$$Q_{con}(\Omega^{(k)}) - Q_{con}(\Omega^*) \leq \frac{2\alpha L \|\Omega^{(0)} - \Omega^*\|_F^2}{(k+1)^2}, \tag{20}$$

*where $\alpha = 1$ for the constant step size setting and $\alpha = c$ for the backtracking step size setting.*

Hence, CONCORD-ISTA and CONCORD-FISTA converge at the rates of $\mathcal{O}(1/k)$ and $\mathcal{O}(1/k^2)$ for the $k^{th}$ iteration.

## 4 Implementation & Numerical Experiments

In this section, we outline algorithm implementation details and present results of our comprehensive numerical evaluation. Section 4.1 gives performance comparisons from using synthetic multivariate Gaussian datasets. These datasets are generated from a wide range of sample sizes ($n$) and dimensionality ($p$). Additionally, convergence of CONCORD-ISTA and CONCORD-FISTA will be illustrated. Section 4.2 has timing results from analyzing a real breast cancer dataset with outliers. Comparisons are made to the coordinate-wise CONCORD implementation in gconcord package for R available at http://cran.r-project.org/web/packages/gconcord/.

For implementing the proposed algorithms, we can take advantage of existing linear algebra libraries. Most of the numerical computations in Algorithms 1 and 2 are linear algebra operations, and, unlike the sequential coordinate-wise CONCORD algorithm, CONCORD-ISTA and CONCORD-FISTA implementations can solve increasingly larger problems as more and more scalable and efficient linear algebra libraries are made available. For this work, we opted to using Eigen library [11] for its sparse linear algebra routines written in C++. Algorithms 1 and 2 were also written in C++ then interfaced to R for testing. Table 1 gives names for various CONCORD-ISTA and CONCORD-FISTA versions using different initial step size choices.

### 4.1 Synthetic Datasets

Synthetic datasets were generated from true sparse positive random $\Omega$ matrices of three sizes: $p = \{1000, 3000, 5000\}$. Instances of random matrices used here consist of 4995, 14985 and 24975 non-zeros, corresponding to 1%, 0.33% and 0.20% edge densities, respectively. For each $p$, Gaussian and t-distributed datasets of sizes $n = \{0.25p, 0.75p, 1.25p\}$ were used as inputs. The initial guess, $\Omega^{(0)}$, and the convergence criteria was matched to those of coordinate-wise CONCORD implementation. Highlights of the results are summarized below, and the complete set of comparisons are given in Supplementary materials Section A.

For normally distributed synthetic datasets, our experiments indicate that two variations of the CONCORD-ISTA method show little performance difference. However, ccista_0 was marginally faster in our tests. On the other hand, ccfista_1 variation of CONCORD-FISTA that uses $\tau_{(k+1,0)} = \tau_k$ as initial step size was significantly faster than ccfista_0. Table 2 gives actual running times for the two best performing algorithms, ccista_0 and ccfista_1, against the coordinate-wise concord. As $p$ and $n$ increase ccista_0 performs very well. For smaller $n$ and $\lambda$, coordinate-wise concord performs well (more in Supplemental section A). This can be attributed to $\min(\mathcal{O}(np^2), \mathcal{O}(p^3))$ computational complexity of coordinate-wise CONCORD [6], and the sparse linear algebra routines used in CONCORD-ISTA and CONCORD-FISTA implementations slowing down as the number of non-zero elements in $\Omega$ increases. On the other hand, for large $n$ fraction ($n = 1.25p$), the proposed methods ccista_0 and ccfista_1 are significantly faster than coordinate-wise concord. In particular, when $p = 5000$ and $n = 6250$, the speed-up of ccista_0 can be as much as 150 times over coordinate-wise concord. Also, for t-distributed synthetic datasets, ccista_0 is generally fastest, especially when $n$ and $p$ are both large.

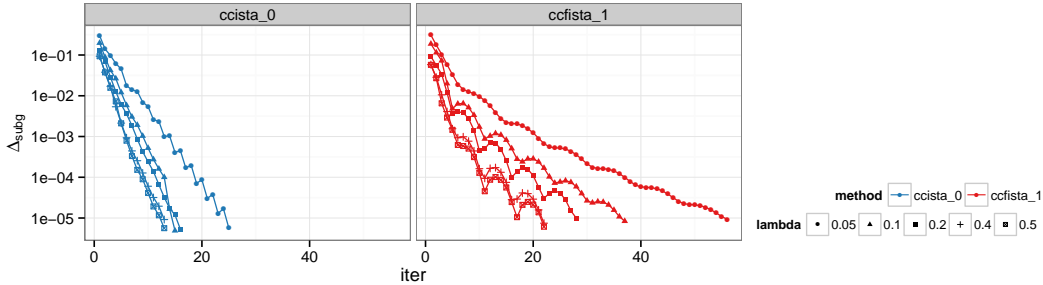

Figure 1: Convergence of CONCORD-ISTA and CONCORD-FISTA for threshold $\Delta_{\mathrm{subg}} < 10^{-5}$

When a good initial guess $\Omega^{(0)}$ is available, warm-starting `cc_ista_0` and `cc_fista_0` algorithms substantially shortens the running times. Simulations with Gaussian datasets indicate the running times can be shortened by, on average, as much as 60%. Complete simulation results are given in the Supplemental Section A.6.

Convergence behavior of CONCORD-ISTA and CONCORD-FISTA methods is shown in Figure 1. The best performing algorithms `ccista_0` and `ccfista_1` are shown. The vertical axis is the subgradient $\Delta_{\mathrm{subg}}$ (See Algorithms 1, 2). Plots show that `ccista_0` seems to converge at a constant rate much faster than `ccfista_1` that appears to slow down after a few initial iterations. While the theoretical convergence results from section 3 prove convergence rates of $\mathcal{O}(1/k)$ and $\mathcal{O}(1/k^2)$ for CONCORD-ISTA and CONCORD-FISTA, in practice, `ccista_0` with constant step size performed the fastest for the tests in this section.

## 4.2 Real Data

Real datasets arising from various physical and biological sciences often are not multivariate Gaussian and can have outliers. Hence, convergence characteristic may be different on such datasets. In this section, the performance of proposed methods are assessed on a breast cancer dataset [12]. This dataset contains expression levels of 24481 genes on 266 patients with breast cancer. Following the approach in Khare et al. [6], the number of genes are reduced by utilizing clinical information that is provided together with the microarray expression dataset. In particular, survival analysis via univariate Cox regression with patient survival times is used to select a subset of genes closely associated with breast cancer. A choice of p-value $< 0.03$ yields a reduced dataset with $p = 4433$ genes.

Often times, graphical model selection algorithms are applied in a non-Gaussian and $n \ll p$ setting such as the case here. In this $n \ll p$ setting, coordinate-wise CONCORD algorithm is especially fast due to its computational complexity $\mathcal{O}(np^2)$. However, even in this setting, the newly proposed methods `ccista_0`, `ccista_1`, and `ccfista_1` perform competitively to, or often better than, `concord` as illustrated in Table 3. On this real dataset, `ccista_1` performed the fastest whereas `ccista_0` was the fastest on synthetic datasets.

## 5 Conclusion

The Gaussian graphical model estimation or inverse covariance estimation has seen tremendous advances in the past few years. In this paper we propose using proximal gradient methods to solve the general non-Gaussian sparse inverse covariance estimation problem. Rates of convergence were established for the CONCORD-ISTA and CONCORD-FISTA algorithms. Coordinate-wise minimization has been the standard approach to this problem thus far, and we provide numerical results comparing CONCORD-ISTA/FISTA and coordinate-wise minimization. We demonstrate that CONCORD-ISTA outperforms coordinate-wise in general, and in high dimensional settings CONCORD-ISTA can outperform coordinate-wise optimization by orders of magnitude. The methodology is also tested on real data sets. We undertake a comprehensive treatment of the problem by also examining the dual formulation and consider methods to maximize the dual objective. We note that efforts similar to ours for the Gaussian case has appeared in not one, but several NIPS and other publications. Our approach on the other hand gives a complete and thorough treatment of the non-Gaussian partial correlation graph estimation problem, all in this one self-contained paper.

Table 1: Naming convention for step size variations

| Variation | concord | ccista_0 | ccista_1 | ccfista_0 | ccfista_1 |
|---|---|---|---|---|---|
| **Method** | Coordinatewise | ISTA | ISTA | FISTA | FISTA |
| **Initial step** | - | Constant | Barzilai-Borwein | Constant | $\tau_k$ |

Table 2: Timing comparison of concord and proposed methods: ccista_0 and ccfista_1.

| p | n | $\lambda$ | NZ% | concord iter | concord seconds | ccista_0 iter | ccista_0 seconds | ccfista_1 iter | ccfista_1 seconds |
|---|---|---|---|---|---|---|---|---|---|
| 1000 | 250 | 0.150 | 1.52 | 9 | 3.2 | 13 | **1.8** | 20 | 3.3 |
| | | 0.163 | 0.99 | 9 | 2.6 | 18 | **2.0** | 26 | 3.3 |
| | | 0.300 | 0.05 | 9 | 2.6 | 15 | **1.2** | 23 | 2.7 |
| | 750 | 0.090 | 1.50 | 9 | 8.9 | 11 | **1.4** | 17 | 2.5 |
| | | 0.103 | 0.76 | 9 | 8.4 | 15 | **1.6** | 24 | 3.3 |
| | | 0.163 | 0.23 | 9 | 8.0 | 15 | **1.6** | 24 | 2.8 |
| | 1250 | 0.071 | 1.41 | 9 | 41.3 | 10 | **1.4** | 17 | 2.9 |
| | | 0.077 | 0.97 | 9 | 40.5 | 15 | **1.7** | 24 | 3.3 |
| | | 0.163 | 0.23 | 9 | 43.8 | 13 | **1.2** | 23 | 2.8 |
| 3000 | 750 | 0.090 | 1.10 | 17 | 147.4 | 20 | **32.4** | 25 | 53.2 |
| | | 0.103 | 0.47 | 17 | 182.4 | 28 | **36.0** | 35 | 60.1 |
| | | 0.163 | 0.08 | 16 | 160.1 | 28 | **28.3** | 26 | 39.9 |
| | 2250 | 0.053 | 1.07 | 16 | 388.3 | 17 | **28.5** | 17 | 39.6 |
| | | 0.059 | 0.56 | 16 | 435.0 | 28 | **38.5** | 26 | 61.9 |
| | | 0.090 | 0.16 | 16 | 379.4 | 16 | **19.9** | 15 | 23.6 |
| | 3750 | 0.040 | 1.28 | 16 | 2854.2 | 17 | **33.0** | 17 | 47.3 |
| | | 0.053 | 0.28 | 16 | 2921.5 | 15 | **23.5** | 16 | 31.4 |
| | | 0.163 | 0.07 | 15 | 2780.5 | 25 | **35.1** | 32 | 56.1 |
| 5000 | 1250 | 0.066 | 1.42 | 17 | 832.7 | 32 | **193.9** | 37 | 379.2 |
| | | 0.077 | 0.53 | 17 | 674.7 | 30 | **121.4** | 35 | 265.8 |
| | | 0.103 | 0.10 | 17 | 667.6 | 27 | **81.2** | 33 | 163.0 |
| | 3750 | 0.039 | 1.36 | 17 | 2102.8 | 18 | **113.0** | 17 | 176.3 |
| | | 0.049 | 0.31 | 17 | 1826.6 | 16 | **73.4** | 17 | 107.4 |
| | | 0.077 | 0.10 | 17 | 2094.7 | 29 | **95.8** | 33 | 178.1 |
| | 6250 | 0.039 | 0.27 | 17 | 15629.3 | 17 | **93.9** | 17 | 130.0 |
| | | 0.077 | 0.10 | 17 | 15671.1 | 27 | **101.0** | 25 | 123.9 |
| | | 0.163 | 0.04 | 16 | 14787.8 | 26 | **97.3** | 34 | 173.7 |

Table 3: Running time comparison on breast cancer dataset

| $\lambda$ | NZ% | concord iter | concord sec | ccista_0 iter | ccista_0 sec | ccista_1 iter | ccista_1 sec | ccfista_0 iter | ccfista_0 sec | ccfista_1 iter | ccfista_1 sec |
|---|---|---|---|---|---|---|---|---|---|---|---|
| 0.450 | 0.110 | 80 | 724.5 | 132 | 686.7 | 123 | **504.0** | 250 | 10870.3 | 201 | 672.6 |
| 0.451 | 0.109 | 80 | 664.2 | 129 | 669.2 | 112 | **457.0** | 216 | 7867.2 | 199 | 662.9 |
| 0.454 | 0.106 | 80 | 690.3 | 130 | 686.2 | 81 | **352.9** | 213 | 7704.2 | 198 | 677.8 |
| 0.462 | 0.101 | 79 | 671.6 | 125 | 640.4 | 109 | **447.1** | 214 | 7978.4 | 196 | 646.3 |
| 0.478 | 0.088 | 77 | 663.3 | 117 | 558.6 | 87 | **337.9** | 202 | 6913.1 | 197 | 609.0 |
| 0.515 | 0.063 | 63 | 600.6 | 104 | 466.0 | 75 | **282.4** | 276 | 9706.9 | 184 | 542.0 |
| 0.602 | 0.027 | 46 | 383.5 | 80 | 308.0 | 66 | **229.7** | 172 | 4685.2 | 152 | 409.1 |
| 0.800 | 0.002 | 24 | 193.6 | 45 | 133.8 | 32 | **92.2** | 74 | 1077.2 | 70 | 169.8 |

**Acknowledgments**: S.O., O.D. and B.R. were supported in part by the National Science Foundation under grants DMS-0906392, DMS-CMG 1025465, AGS-1003823, DMS-1106642, DMS CAREER-1352656 and grants DARPA-YFAN66001-111-4131 and SMC-DBNKY. K.K was partially supported by NSF grant DMS-1106084. S.O. was supported also in part by the Laboratory Directed Research and Development Program of Lawrence Berkeley National Laboratory under U.S. Department of Energy Contract No. DE-AC02-05CH11231.

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
