[Supplementary Material]



## A  Timing comparison

### A.1  Median Speed-up

Table 4: Median speed-up ratio over CONCORD method and (standard deviation).

| p | n | Relative to `concord` | |
| --- | --- | --- | --- |
| | | `ccista_0` | `ccfista_1` |
| 1000 | 250 | 0.6 ( 0.7) | 0.4 ( 0.3) |
| 1000 | 750 | 3.4 ( 1.8) | 1.9 ( 0.9) |
| 1000 | 1250 | 23.1 ( 5.7) | 12.0 ( 3.5) |
| 3000 | 750 | 2.7 ( 2.1) | 1.9 ( 1.6) |
| 3000 | 2250 | 12.8 ( 1.6) | 8.8 ( 2.2) |
| 3000 | 3750 | 81.9 ( 6.6) | 58.2 ( 8.7) |
| 5000 | 1250 | 5.6 ( 3.2) | 3.0 ( 1.8) |
| 5000 | 3750 | 21.1 ( 2.6) | 13.5 ( 2.6) |
| 5000 | 6250 | 145.8 ( 6.6) | 110.1 (16.4) |

### A.2  Comparison among CONCORD-ISTA and CONCORD-FISTA variations

Figure 2: Timings of CONCORD-ISTA (top) and CONCORD-FISTA (bottom) variations for sample sizes $n = \{0.25p, 0.75p, 1.25p\}$

## A.3 Comparison with CONCORD algorithm

Figure 3: Timing of best CONCORD-ISTA and CONCORD-FISTA variations against CONCORD for sample sizes $n = \{0.25p, 0.75p, 1.25p\}$.

## A.4 Running times (Gaussian data)

Table 5: $p = 1000$, true non-zero fraction (nzf) of 1%

| $p$ | $n$ | $\lambda$ | nzf (%) | concord | | ccista_0 | | ccfista_1 | |
|---|---|---|---|---|---|---|---|---|---|
| | | | | iter | seconds | iter | seconds | iter | seconds |
| 1000 | 250 | 0.058 | 20.46 | 22 | 7.74 | 58 | 33.05 | 64 | 52.04 |
| 1000 | 250 | 0.059 | 20.04 | 21 | 7.70 | 55 | 24.84 | 63 | 36.13 |
| 1000 | 250 | 0.061 | 19.10 | 20 | 6.74 | 56 | 35.06 | 60 | 31.71 |
| 1000 | 250 | 0.066 | 17.05 | 17 | 5.83 | 45 | 26.86 | 54 | 28.59 |
| 1000 | 250 | 0.077 | 13.00 | 13 | 4.21 | 34 | 17.99 | 44 | 18.90 |
| 1000 | 250 | 0.103 | 6.59 | 10 | 3.11 | 22 | 6.01 | 33 | 9.44 |
| 1000 | 250 | 0.163 | 0.99 | 9 | 2.61 | 18 | 1.98 | 26 | 3.31 |
| 1000 | 250 | 0.300 | 0.05 | 9 | 2.58 | 15 | 1.23 | 23 | 2.67 |
| 1000 | 750 | 0.058 | 8.99 | 10 | 9.96 | 20 | 4.56 | 28 | 12.44 |
| 1000 | 750 | 0.059 | 8.56 | 10 | 9.86 | 20 | 5.19 | 28 | 9.86 |
| 1000 | 750 | 0.061 | 7.64 | 10 | 9.97 | 20 | 5.41 | 28 | 7.96 |
| 1000 | 750 | 0.066 | 5.86 | 10 | 10.45 | 20 | 4.01 | 27 | 6.96 |
| 1000 | 750 | 0.077 | 3.09 | 9 | 8.37 | 16 | 3.53 | 25 | 4.84 |
| 1000 | 750 | 0.103 | 0.76 | 9 | 8.40 | 15 | 1.58 | 24 | 3.26 |
| 1000 | 750 | 0.163 | 0.23 | 9 | 8.00 | 15 | 1.57 | 24 | 2.80 |
| 1000 | 750 | 0.300 | 0.04 | 8 | 6.96 | 13 | 1.13 | 18 | 2.20 |
| 1000 | 1250 | 0.058 | 3.69 | 9 | 44.21 | 15 | 2.54 | 24 | 5.29 |
| 1000 | 1250 | 0.059 | 3.43 | 9 | 44.25 | 16 | 2.49 | 24 | 5.00 |
| 1000 | 1250 | 0.061 | 2.91 | 9 | 43.84 | 16 | 2.36 | 24 | 5.38 |
| 1000 | 1250 | 0.066 | 2.03 | 9 | 44.15 | 14 | 1.79 | 24 | 4.09 |
| 1000 | 1250 | 0.077 | 0.97 | 9 | 40.50 | 15 | 1.65 | 24 | 3.34 |
| 1000 | 1250 | 0.103 | 0.44 | 9 | 44.16 | 15 | 1.93 | 24 | 3.02 |
| 1000 | 1250 | 0.163 | 0.23 | 9 | 43.84 | 13 | 1.25 | 23 | 2.75 |
| 1000 | 1250 | 0.300 | 0.04 | 8 | 35.99 | 13 | 1.53 | 17 | 2.13 |

Table 6: $p = 3000$, true non-zero fraction (nzf) of 0.33%

| $p$ | $n$ | $\lambda$ | nzf (%) | concord | | ccista_0 | | ccfista_1 | |
|---|---|---|---|---|---|---|---|---|---|
| | | | | iter | seconds | iter | seconds | iter | seconds |
| 3000 | 750 | 0.077 | 2.42 | 18 | 190.00 | 36 | 85.81 | 31 | 135.13 |
| 3000 | 750 | 0.103 | 0.47 | 17 | 182.36 | 28 | 36.00 | 35 | 60.13 |
| 3000 | 750 | 0.163 | 0.08 | 16 | 160.13 | 28 | 28.29 | 26 | 39.94 |
| 3000 | 750 | 0.300 | 0.01 | 15 | 147.07 | 25 | 29.67 | 23 | 34.80 |
| 3000 | 2250 | 0.058 | 0.61 | 16 | 433.05 | 27 | 36.63 | 26 | 62.26 |
| 3000 | 2250 | 0.059 | 0.56 | 16 | 434.96 | 28 | 38.50 | 26 | 61.90 |
| 3000 | 2250 | 0.061 | 0.45 | 16 | 425.58 | 28 | 36.75 | 26 | 50.02 |
| 3000 | 2250 | 0.066 | 0.30 | 16 | 400.08 | 28 | 34.55 | 34 | 66.10 |
| 3000 | 2250 | 0.077 | 0.19 | 16 | 464.53 | 28 | 33.57 | 32 | 55.90 |
| 3000 | 2250 | 0.103 | 0.14 | 16 | 462.08 | 28 | 37.39 | 24 | 41.50 |
| 3000 | 2250 | 0.163 | 0.07 | 15 | 420.28 | 26 | 29.57 | 25 | 42.17 |
| 3000 | 2250 | 0.300 | 0.01 | 14 | 391.94 | 22 | 25.06 | 22 | 31.20 |
| 3000 | 3750 | 0.058 | 0.22 | 16 | 2837.71 | 27 | 32.61 | 24 | 41.36 |
| 3000 | 3750 | 0.059 | 0.21 | 16 | 2993.98 | 27 | 33.59 | 24 | 50.58 |
| 3000 | 3750 | 0.061 | 0.20 | 16 | 2826.17 | 27 | 33.06 | 24 | 45.75 |
| 3000 | 3750 | 0.066 | 0.19 | 16 | 2805.85 | 27 | 36.94 | 31 | 57.06 |
| 3000 | 3750 | 0.077 | 0.17 | 15 | 2792.55 | 26 | 36.61 | 31 | 48.96 |
| 3000 | 3750 | 0.103 | 0.14 | 15 | 2649.75 | 26 | 36.43 | 31 | 53.95 |
| 3000 | 3750 | 0.163 | 0.07 | 15 | 2780.53 | 25 | 35.12 | 32 | 56.06 |
| 3000 | 3750 | 0.300 | 0.01 | 13 | 2406.49 | 22 | 26.91 | 22 | 33.90 |

Table 7: $p = 5000$, true non-zero fraction (nzf) of 0.20%

| $p$ | $n$ | $\lambda$ | nzf (%) | concord | | ccista_0 | | ccfista_1 | |
|---|---|---|---|---|---|---|---|---|---|
| | | | | iter | seconds | iter | seconds | iter | seconds |
| 5000 | 1250 | 0.058 | 2.71 | 18 | 757.67 | 38 | 408.49 | 40 | 547.93 |
| 5000 | 1250 | 0.059 | 2.52 | 18 | 903.05 | 37 | 393.77 | 40 | 681.49 |
| 5000 | 1250 | 0.061 | 2.13 | 18 | 892.30 | 36 | 272.03 | 40 | 604.35 |
| 5000 | 1250 | 0.066 | 1.42 | 17 | 832.68 | 32 | 193.88 | 37 | 379.23 |
| 5000 | 1250 | 0.077 | 0.53 | 17 | 674.71 | 30 | 121.39 | 35 | 265.84 |
| 5000 | 1250 | 0.103 | 0.10 | 17 | 667.62 | 27 | 81.21 | 33 | 163.00 |
| 5000 | 1250 | 0.163 | 0.05 | 16 | 719.81 | 25 | 71.23 | 34 | 147.53 |
| 5000 | 1250 | 0.300 | 0.01 | 14 | 626.20 | 25 | 69.71 | 30 | 105.65 |
| 5000 | 3750 | 0.058 | 0.14 | 17 | 2324.54 | 29 | 99.50 | 35 | 165.12 |
| 5000 | 3750 | 0.059 | 0.13 | 17 | 1965.36 | 29 | 111.53 | 35 | 189.05 |
| 5000 | 3750 | 0.061 | 0.13 | 17 | 1967.39 | 29 | 114.72 | 35 | 186.34 |
| 5000 | 3750 | 0.066 | 0.11 | 17 | 2183.90 | 29 | 98.54 | 25 | 121.39 |
| 5000 | 3750 | 0.077 | 0.10 | 17 | 2094.73 | 29 | 95.84 | 33 | 178.13 |
| 5000 | 3750 | 0.103 | 0.08 | 16 | 1780.97 | 26 | 88.29 | 32 | 141.14 |
| 5000 | 3750 | 0.163 | 0.04 | 16 | 2021.49 | 25 | 82.88 | 33 | 133.36 |
| 5000 | 3750 | 0.300 | 0.01 | 14 | 1767.63 | 24 | 78.77 | 30 | 117.03 |
| 5000 | 6250 | 0.058 | 0.12 | 17 | 15698.02 | 27 | 113.65 | 25 | 150.95 |
| 5000 | 6250 | 0.059 | 0.12 | 17 | 16221.44 | 27 | 115.35 | 25 | 130.19 |
| 5000 | 6250 | 0.061 | 0.11 | 17 | 15698.53 | 27 | 103.06 | 25 | 132.57 |
| 5000 | 6250 | 0.066 | 0.11 | 17 | 16220.33 | 27 | 111.75 | 25 | 129.70 |
| 5000 | 6250 | 0.077 | 0.10 | 17 | 15671.14 | 27 | 101.03 | 25 | 123.92 |
| 5000 | 6250 | 0.103 | 0.08 | 17 | 15600.83 | 26 | 112.48 | 33 | 144.42 |
| 5000 | 6250 | 0.163 | 0.04 | 16 | 14787.78 | 26 | 97.33 | 34 | 173.66 |
| 5000 | 6250 | 0.300 | 0.01 | 14 | 13287.76 | 24 | 91.84 | 30 | 149.70 |

## A.5 Running times (t data)

Table 8: $p = 1000$, true non-zero fraction (nzf) of 1%

| p | n | $\lambda$ | NZ% | concord | | ccista_0 | | ccfista_1 | |
|---|---|---|---|---|---|---|---|---|---|
| | | | | iter | seconds | iter | seconds | iter | seconds |
| 1000 | 250 | 0.236 | 1.48 | 33 | 12.7 | 62 | 6.9 | 214 | 43.1 |
| 1000 | 250 | 0.267 | 1.05 | 31 | 17.4 | 55 | 5.5 | 201 | 39.4 |
| 1000 | 250 | 0.305 | 0.67 | 28 | 12.1 | 49 | 5.4 | 188 | 30.0 |
| 1000 | 250 | 0.350 | 0.38 | 23 | 12.8 | 44 | 4.3 | 99 | 16.6 |
| 1000 | 750 | 0.128 | 2.99 | 14 | 28.3 | 46 | 8.1 | 67 | 12.2 |
| 1000 | 750 | 0.132 | 2.77 | 14 | 30.0 | 45 | 4.5 | 66 | 11.8 |
| 1000 | 750 | 0.137 | 2.47 | 13 | 22.9 | 45 | 5.2 | 65 | 11.2 |
| 1000 | 750 | 0.146 | 2.05 | 13 | 24.2 | 44 | 4.2 | 64 | 14.3 |
| 1000 | 750 | 0.159 | 1.55 | 12 | 24.6 | 42 | 5.3 | 63 | 12.3 |
| 1000 | 750 | 0.178 | 1.02 | 11 | 17.0 | 40 | 3.4 | 59 | 6.1 |
| 1000 | 750 | 0.207 | 0.52 | 9 | 10.0 | 39 | 4.5 | 33 | 3.2 |
| 1000 | 750 | 0.250 | 0.19 | 9 | 13.9 | 38 | 2.9 | 27 | 3.1 |
| 1000 | 1250 | 0.105 | 2.96 | 11 | 99.2 | 41 | 7.6 | 56 | 9.9 |
| 1000 | 1250 | 0.107 | 2.76 | 11 | 92.2 | 41 | 4.5 | 35 | 5.8 |
| 1000 | 1250 | 0.111 | 2.50 | 10 | 91.8 | 40 | 6.2 | 35 | 5.0 |
| 1000 | 1250 | 0.116 | 2.15 | 10 | 96.6 | 40 | 6.1 | 34 | 5.1 |
| 1000 | 1250 | 0.124 | 1.70 | 9 | 85.2 | 39 | 6.1 | 34 | 4.6 |
| 1000 | 1250 | 0.136 | 1.19 | 9 | 88.3 | 38 | 5.2 | 33 | 4.0 |
| 1000 | 1250 | 0.153 | 0.71 | 9 | 75.2 | 38 | 3.1 | 32 | 6.5 |
| 1000 | 1250 | 0.180 | 0.34 | 9 | 80.1 | 37 | 3.2 | 32 | 4.8 |

Table 9: $p = 3000$, true non-zero fraction (nzf) of 0.33%

| p | n | $\lambda$ | NZ% | concord | | ccista_0 | | ccfista_1 | |
|---|---|---|---|---|---|---|---|---|---|
| | | | | iter | seconds | iter | seconds | iter | seconds |
| 3000 | 750 | 0.145 | 1.21 | 74 | 719.2 | 136 | 1185.8 | 1000 | 5625.5 |
| 3000 | 750 | 0.171 | 0.79 | 67 | 620.5 | 133 | 820.6 | 999 | 4751.7 |
| 3000 | 750 | 0.210 | 0.43 | 56 | 535.3 | 119 | 546.8 | 947 | 3864.8 |
| 3000 | 2250 | 0.090 | 1.90 | 26 | 679.4 | 73 | 545.5 | 382 | 2738.1 |
| 3000 | 2250 | 0.093 | 1.72 | 26 | 662.4 | 74 | 546.7 | 379 | 2466.8 |
| 3000 | 2250 | 0.099 | 1.49 | 25 | 635.0 | 69 | 298.9 | 208 | 1314.1 |
| 3000 | 2250 | 0.106 | 1.19 | 24 | 589.5 | 69 | 283.1 | 205 | 1068.7 |
| 3000 | 2250 | 0.118 | 0.86 | 23 | 662.9 | 69 | 260.8 | 201 | 985.9 |
| 3000 | 2250 | 0.135 | 0.53 | 22 | 624.8 | 62 | 241.6 | 197 | 745.4 |
| 3000 | 2250 | 0.161 | 0.26 | 20 | 502.4 | 63 | 183.0 | 112 | 319.8 |
| 3000 | 2250 | 0.200 | 0.10 | 15 | 421.2 | 57 | 217.7 | 95 | 284.8 |
| 3000 | 3750 | 0.080 | 1.85 | 19 | 3255.9 | 69 | 421.2 | 211 | 1374.9 |
| 3000 | 3750 | 0.081 | 1.78 | 19 | 3315.8 | 70 | 501.2 | 210 | 1389.7 |
| 3000 | 3750 | 0.083 | 1.67 | 19 | 2992.7 | 71 | 509.0 | 209 | 1429.6 |
| 3000 | 3750 | 0.086 | 1.52 | 19 | 3349.6 | 68 | 428.7 | 208 | 1441.0 |
| 3000 | 3750 | 0.090 | 1.32 | 18 | 3293.7 | 67 | 438.7 | 206 | 1229.1 |
| 3000 | 3750 | 0.096 | 1.07 | 18 | 3247.3 | 67 | 369.5 | 204 | 1248.7 |
| 3000 | 3750 | 0.106 | 0.79 | 17 | 3037.3 | 66 | 194.5 | 117 | 522.7 |
| 3000 | 3750 | 0.120 | 0.50 | 16 | 2963.8 | 61 | 170.1 | 114 | 491.4 |

Table 10: $p = 5000$, true non-zero fraction (nzf) of 0.20%

| p | n | $\lambda$ | NZ% | concord | | ccista_0 | | ccfista_1 | |
|---|---|---|---|---|---|---|---|---|---|
| | | | | iter | seconds | iter | seconds | iter | seconds |
| 5000 | 1250 | 0.114 | 1.13 | 21 | 832.3 | 73 | 1088.0 | 218 | 3727.9 |
| 5000 | 1250 | 0.123 | 0.92 | 20 | 812.5 | 68 | 847.4 | 213 | 3427.7 |
| 5000 | 1250 | 0.136 | 0.67 | 18 | 808.9 | 67 | 584.0 | 208 | 3083.0 |
| 5000 | 1250 | 0.155 | 0.41 | 17 | 737.8 | 65 | 588.5 | 117 | 1733.9 |
| 5000 | 3750 | 0.090 | 0.78 | 24 | 2935.1 | 73 | 1148.9 | 355 | 5327.0 |
| 5000 | 3750 | 0.091 | 0.76 | 24 | 2881.4 | 70 | 1153.8 | 355 | 5250.8 |
| 5000 | 3750 | 0.092 | 0.74 | 24 | 3024.6 | 71 | 1054.1 | 355 | 5673.5 |
| 5000 | 3750 | 0.093 | 0.70 | 24 | 2989.9 | 72 | 1050.1 | 354 | 5324.6 |
| 5000 | 3750 | 0.095 | 0.64 | 24 | 3039.3 | 72 | 1005.6 | 203 | 2984.2 |
| 5000 | 3750 | 0.098 | 0.57 | 23 | 2814.4 | 66 | 878.8 | 202 | 2713.0 |
| 5000 | 3750 | 0.103 | 0.47 | 21 | 2468.2 | 66 | 720.9 | 200 | 2437.7 |
| 5000 | 3750 | 0.110 | 0.36 | 21 | 2549.3 | 63 | 611.6 | 198 | 2429.0 |
| 5000 | 6250 | 0.080 | 0.88 | 37 | 34226.4 | 85 | 1844.1 | 615 | 11091.0 |
| 5000 | 6250 | 0.080 | 0.88 | 37 | 33600.1 | 89 | 2018.7 | 615 | 10815.6 |
| 5000 | 6250 | 0.080 | 0.87 | 37 | 34372.5 | 87 | 1790.7 | 614 | 10930.0 |
| 5000 | 6250 | 0.081 | 0.86 | 37 | 34582.0 | 86 | 1897.3 | 614 | 10978.6 |
| 5000 | 6250 | 0.081 | 0.84 | 37 | 33310.6 | 86 | 1613.2 | 614 | 10635.1 |
| 5000 | 6250 | 0.082 | 0.82 | 37 | 34511.9 | 85 | 1469.4 | 613 | 10689.6 |
| 5000 | 6250 | 0.083 | 0.78 | 37 | 34432.1 | 88 | 1911.8 | 612 | 10615.6 |
| 5000 | 6250 | 0.085 | 0.74 | 37 | 34366.0 | 88 | 1855.4 | 609 | 10050.4 |

## A.6 Warm start: speed-up of average compute times

Fix $p$ and $n$. For each penalty parameter in a decreasing sequence $\Lambda = \lambda_1, \lambda_2, \ldots \lambda_l$, a CONCORD estimator $\hat{\Omega}_{\lambda_i}$ is computed by either cold-starting from an identity matrix or warm-starting from the previously computed $\hat{\Omega}_{\lambda_{i-1}}$. The mean over ratios of elapsed times $t^{\text{warm}}_{\lambda_i}/t^{\text{cold}}_{\lambda_i}$ for $i = 1, 2, \ldots, l$ , are presented as comparison in the Table 11. Note that $t^{\text{cold}}_{\lambda_i}$ denotes running time when cold-starting, and $t^{\text{warm}}_{\lambda_i}$ denotes running time when warm-starting. Also, $\tau_{(0,0)} = 0.25$ and the data used is generated from a normal distribution in this section.

Table 11: Mean speed-up of warm starting from a nearby solution

| p | | 1000 | | | 3000 | | | 5000 | |
|---|---|---|---|---|---|---|---|---|---|
| n | 250 | 750 | 1250 | 750 | 2250 | 3750 | 1250 | 3750 | 6250 |
| ccista_0 | 0.595 | 0.63 | 0.569 | 0.396 | 0.429 | 0.457 | 0.47 | 0.37 | 0.463 |
| ccfista_1 | 0.541 | 0.652 | 0.578 | 0.577 | 0.53 | 0.616 | 0.567 | 0.481 | 0.633 |

## A.7 Running times (warm start)

Table 12: $p = 1000$, true non-zero fraction (nzf) of 1%

| | | | | ccista_0 | | | | | |
|---|---|---|---|---|---|---|---|---|---|
| p | n | $\lambda$ | NZ% | Cold | | Warm | | Warm/Cold | |
| | | | | iter | seconds | iter | seconds | iter (ratio) | seconds (ratio) |
| 1000 | 250 | 0.25 | 0.098 | 36 | 2.47 | 36 | 2.65 | 1.0 | 1.07 |
| 1000 | 250 | 0.21 | 0.189 | 37 | 2.62 | 25 | 1.51 | 0.7 | 0.58 |
| 1000 | 250 | 0.19 | 0.381 | 37 | 2.61 | 24 | 1.55 | 0.6 | 0.59 |
| 1000 | 250 | 0.18 | 0.641 | 38 | 2.83 | 24 | 1.69 | 0.6 | 0.60 |
| 1000 | 250 | 0.16 | 0.943 | 38 | 2.98 | 24 | 1.72 | 0.6 | 0.58 |
| 1000 | 250 | 0.16 | 1.195 | 39 | 3.79 | 23 | 1.79 | 0.6 | 0.47 |
| 1000 | 250 | 0.15 | 1.387 | 39 | 3.53 | 22 | 1.84 | 0.6 | 0.52 |
| 1000 | 250 | 0.15 | 1.540 | 40 | 5.42 | 20 | 1.86 | 0.5 | 0.34 |

| p | n | λ | NZ% | Cold iter | Cold seconds | Warm iter | Warm seconds | Warm/Cold iter (ratio) | Warm/Cold seconds (ratio) |
|---|---|---|---|---|---|---|---|---|---|
| 1000 | 750 | 0.22 | 0.108 | 36 | 2.32 | 28 | 2.35 | 0.8 | 1.01 |
| 1000 | 750 | 0.17 | 0.204 | 37 | 2.45 | 25 | 1.68 | 0.7 | 0.69 |
| 1000 | 750 | 0.14 | 0.289 | 37 | 2.71 | 24 | 1.66 | 0.6 | 0.61 |
| 1000 | 750 | 0.12 | 0.404 | 38 | 2.84 | 23 | 1.68 | 0.6 | 0.59 |
| 1000 | 750 | 0.11 | 0.592 | 38 | 3.12 | 22 | 1.94 | 0.6 | 0.62 |
| 1000 | 750 | 0.10 | 0.883 | 39 | 3.22 | 21 | 1.69 | 0.5 | 0.52 |
| 1000 | 750 | 0.09 | 1.216 | 39 | 3.45 | 20 | 1.70 | 0.5 | 0.49 |
| 1000 | 750 | 0.09 | 1.506 | 39 | 4.07 | 19 | 2.04 | 0.5 | 0.50 |
| 1000 | 1250 | 0.20 | 0.138 | 35 | 2.52 | 30 | 2.08 | 0.9 | 0.83 |
| 1000 | 1250 | 0.15 | 0.253 | 36 | 2.75 | 25 | 1.80 | 0.7 | 0.65 |
| 1000 | 1250 | 0.12 | 0.346 | 37 | 2.91 | 24 | 1.75 | 0.6 | 0.60 |
| 1000 | 1250 | 0.10 | 0.438 | 37 | 2.91 | 23 | 1.68 | 0.6 | 0.58 |
| 1000 | 1250 | 0.09 | 0.574 | 38 | 3.66 | 22 | 1.67 | 0.6 | 0.46 |
| 1000 | 1250 | 0.08 | 0.832 | 38 | 3.86 | 21 | 1.68 | 0.6 | 0.44 |
| 1000 | 1250 | 0.07 | 1.179 | 38 | 3.36 | 20 | 1.71 | 0.5 | 0.51 |
| 1000 | 1250 | 0.07 | 1.539 | 38 | 3.48 | 19 | 1.72 | 0.5 | 0.49 |

| ccfista_1 | | | | | | | | | |
|---|---|---|---|---|---|---|---|---|---|
| p | n | λ | NZ% | Cold iter | Cold seconds | Warm iter | Warm seconds | Warm/Cold iter (ratio) | Warm/Cold seconds (ratio) |
| 1000 | 250 | 0.25 | 0.098 | 31 | 3.22 | 31 | 2.39 | 1.0 | 0.74 |
| 1000 | 250 | 0.21 | 0.189 | 32 | 2.89 | 19 | 1.61 | 0.6 | 0.56 |
| 1000 | 250 | 0.19 | 0.381 | 32 | 3.13 | 19 | 1.59 | 0.6 | 0.51 |
| 1000 | 250 | 0.18 | 0.641 | 33 | 3.57 | 19 | 2.27 | 0.6 | 0.64 |
| 1000 | 250 | 0.16 | 0.943 | 33 | 3.85 | 19 | 1.88 | 0.6 | 0.49 |
| 1000 | 250 | 0.16 | 1.195 | 34 | 4.39 | 19 | 2.21 | 0.6 | 0.50 |
| 1000 | 250 | 0.15 | 1.387 | 34 | 4.63 | 18 | 2.37 | 0.5 | 0.51 |
| 1000 | 250 | 0.15 | 1.540 | 34 | 4.86 | 13 | 1.82 | 0.4 | 0.37 |
| 1000 | 750 | 0.22 | 0.108 | 31 | 2.37 | 25 | 2.08 | 0.8 | 0.88 |
| 1000 | 750 | 0.17 | 0.204 | 31 | 2.38 | 19 | 2.01 | 0.6 | 0.84 |
| 1000 | 750 | 0.14 | 0.289 | 32 | 2.53 | 18 | 1.75 | 0.6 | 0.69 |
| 1000 | 750 | 0.12 | 0.404 | 32 | 3.10 | 18 | 1.84 | 0.6 | 0.59 |
| 1000 | 750 | 0.11 | 0.592 | 32 | 3.57 | 18 | 2.20 | 0.6 | 0.62 |
| 1000 | 750 | 0.10 | 0.883 | 33 | 4.09 | 18 | 2.65 | 0.5 | 0.65 |
| 1000 | 750 | 0.09 | 1.216 | 33 | 3.77 | 17 | 2.24 | 0.5 | 0.59 |
| 1000 | 750 | 0.09 | 1.506 | 33 | 4.68 | 13 | 1.66 | 0.4 | 0.36 |
| 1000 | 1250 | 0.20 | 0.138 | 31 | 2.88 | 25 | 2.23 | 0.8 | 0.78 |
| 1000 | 1250 | 0.15 | 0.253 | 31 | 3.05 | 23 | 2.08 | 0.7 | 0.68 |
| 1000 | 1250 | 0.12 | 0.346 | 32 | 3.12 | 18 | 1.74 | 0.6 | 0.56 |
| 1000 | 1250 | 0.10 | 0.438 | 32 | 3.26 | 18 | 2.21 | 0.6 | 0.68 |
| 1000 | 1250 | 0.09 | 0.574 | 32 | 3.70 | 18 | 2.22 | 0.6 | 0.60 |
| 1000 | 1250 | 0.08 | 0.832 | 32 | 4.35 | 17 | 1.88 | 0.5 | 0.43 |
| 1000 | 1250 | 0.07 | 1.179 | 32 | 4.38 | 17 | 2.59 | 0.5 | 0.59 |
| 1000 | 1250 | 0.07 | 1.539 | 33 | 5.28 | 12 | 1.64 | 0.4 | 0.31 |

Table 12: $p = 1000$

Table 13: $p = 3000$, true non-zero fraction (nzf) of 0.33%

| ccista_0 | | | | | | | | | |
|---|---|---|---|---|---|---|---|---|---|
| p | n | λ | NZ% | Cold iter | Cold seconds | Warm iter | Warm seconds | Warm/Cold iter (ratio) | Warm/Cold seconds (ratio) |
| 3000 | 750 | 0.16 | 0.081 | 61 | 69.93 | 61 | 54.15 | 1.0 | 0.77 |
| 3000 | 750 | 0.14 | 0.118 | 62 | 75.70 | 26 | 23.93 | 0.4 | 0.32 |
| 3000 | 750 | 0.12 | 0.199 | 63 | 69.40 | 24 | 23.77 | 0.4 | 0.34 |
| 3000 | 750 | 0.11 | 0.359 | 64 | 69.55 | 23 | 27.91 | 0.4 | 0.40 |
| 3000 | 750 | 0.10 | 0.565 | 64 | 77.40 | 22 | 27.72 | 0.3 | 0.36 |
| 3000 | 750 | 0.10 | 0.773 | 65 | 87.29 | 22 | 29.48 | 0.3 | 0.34 |
| 3000 | 750 | 0.09 | 0.956 | 65 | 88.04 | 21 | 27.87 | 0.3 | 0.32 |

| p | n | λ | NZ% | Cold iter | Cold seconds | Warm iter | Warm seconds | Warm/Cold iter (ratio) | Warm/Cold seconds (ratio) |
|---|---|---|---|---|---|---|---|---|---|
| 3000 | 750 | 0.09 | 1.100 | 65 | 100.89 | 20 | 32.62 | 0.3 | 0.32 |
| 3000 | 2250 | 0.13 | 0.107 | 58 | 57.99 | 37 | 40.45 | 0.6 | 0.70 |
| 3000 | 2250 | 0.10 | 0.139 | 60 | 58.61 | 25 | 29.16 | 0.4 | 0.50 |
| 3000 | 2250 | 0.08 | 0.164 | 60 | 62.71 | 24 | 25.49 | 0.4 | 0.41 |
| 3000 | 2250 | 0.07 | 0.214 | 61 | 80.81 | 22 | 25.52 | 0.4 | 0.32 |
| 3000 | 2250 | 0.06 | 0.340 | 61 | 70.90 | 21 | 31.30 | 0.3 | 0.44 |
| 3000 | 2250 | 0.06 | 0.554 | 62 | 79.03 | 20 | 31.01 | 0.3 | 0.39 |
| 3000 | 2250 | 0.06 | 0.816 | 62 | 93.76 | 19 | 31.05 | 0.3 | 0.33 |
| 3000 | 2250 | 0.05 | 1.068 | 62 | 92.50 | 18 | 32.08 | 0.3 | 0.35 |
| 3000 | 3750 | 0.11 | 0.129 | 57 | 59.35 | 36 | 51.36 | 0.6 | 0.87 |
| 3000 | 3750 | 0.09 | 0.160 | 58 | 64.93 | 25 | 31.96 | 0.4 | 0.49 |
| 3000 | 3750 | 0.07 | 0.182 | 59 | 74.21 | 23 | 31.19 | 0.4 | 0.42 |
| 3000 | 3750 | 0.06 | 0.222 | 59 | 76.23 | 21 | 28.95 | 0.4 | 0.38 |
| 3000 | 3750 | 0.05 | 0.346 | 60 | 71.07 | 20 | 29.43 | 0.3 | 0.41 |
| 3000 | 3750 | 0.05 | 0.597 | 60 | 77.98 | 19 | 29.11 | 0.3 | 0.37 |
| 3000 | 3750 | 0.04 | 0.936 | 60 | 88.24 | 19 | 29.18 | 0.3 | 0.33 |
| 3000 | 3750 | 0.04 | 1.284 | 60 | 90.95 | 18 | 34.66 | 0.3 | 0.38 |

| | | | | ccfista_1 | | | | | |
|---|---|---|---|---|---|---|---|---|---|
| **p** | **n** | **λ** | **NZ%** | **Cold** | | **Warm** | | **Warm/Cold** | |
| | | | | iter | seconds | iter | seconds | iter (ratio) | seconds (ratio) |
| 3000 | 750 | 0.16 | 0.081 | 34 | 40.35 | 34 | 45.74 | 1.0 | 1.13 |
| 3000 | 750 | 0.14 | 0.118 | 34 | 41.53 | 18 | 22.15 | 0.5 | 0.53 |
| 3000 | 750 | 0.12 | 0.199 | 35 | 49.90 | 18 | 26.05 | 0.5 | 0.52 |
| 3000 | 750 | 0.11 | 0.359 | 35 | 47.69 | 18 | 28.30 | 0.5 | 0.59 |
| 3000 | 750 | 0.10 | 0.565 | 35 | 56.61 | 18 | 29.63 | 0.5 | 0.52 |
| 3000 | 750 | 0.10 | 0.773 | 35 | 61.02 | 18 | 33.02 | 0.5 | 0.54 |
| 3000 | 750 | 0.09 | 0.956 | 35 | 64.17 | 13 | 24.45 | 0.4 | 0.38 |
| 3000 | 750 | 0.09 | 1.101 | 35 | 76.27 | 13 | 29.42 | 0.4 | 0.39 |
| 3000 | 2250 | 0.13 | 0.107 | 33 | 43.32 | 24 | 37.58 | 0.7 | 0.87 |
| 3000 | 2250 | 0.10 | 0.139 | 34 | 43.70 | 18 | 24.62 | 0.5 | 0.56 |
| 3000 | 2250 | 0.08 | 0.164 | 34 | 51.12 | 18 | 30.03 | 0.5 | 0.59 |
| 3000 | 2250 | 0.07 | 0.214 | 34 | 52.91 | 18 | 30.79 | 0.5 | 0.58 |
| 3000 | 2250 | 0.06 | 0.340 | 34 | 51.43 | 12 | 19.57 | 0.4 | 0.38 |
| 3000 | 2250 | 0.06 | 0.554 | 35 | 58.52 | 12 | 26.53 | 0.3 | 0.45 |
| 3000 | 2250 | 0.06 | 0.816 | 35 | 75.48 | 12 | 30.24 | 0.3 | 0.40 |
| 3000 | 2250 | 0.05 | 1.068 | 35 | 68.77 | 12 | 27.62 | 0.3 | 0.40 |
| 3000 | 3750 | 0.11 | 0.129 | 33 | 44.35 | 24 | 47.73 | 0.7 | 1.08 |
| 3000 | 3750 | 0.09 | 0.160 | 33 | 46.90 | 18 | 29.62 | 0.5 | 0.63 |
| 3000 | 3750 | 0.07 | 0.182 | 34 | 54.05 | 18 | 34.39 | 0.5 | 0.64 |
| 3000 | 3750 | 0.06 | 0.222 | 34 | 47.73 | 17 | 33.91 | 0.5 | 0.71 |
| 3000 | 3750 | 0.05 | 0.346 | 34 | 52.95 | 12 | 29.32 | 0.4 | 0.55 |
| 3000 | 3750 | 0.05 | 0.597 | 34 | 62.71 | 12 | 25.86 | 0.4 | 0.41 |
| 3000 | 3750 | 0.04 | 0.936 | 34 | 68.09 | 12 | 32.02 | 0.4 | 0.47 |
| 3000 | 3750 | 0.04 | 1.284 | 34 | 73.68 | 12 | 32.05 | 0.4 | 0.43 |

Table 14: $p = 5000$, true non-zero fraction (nzf) of 0.20%

| | | | | ccista_0 | | | | | |
|---|---|---|---|---|---|---|---|---|---|
| **p** | **n** | **λ** | **NZ%** | **Cold** | | **Warm** | | **Warm/Cold** | |
| | | | | iter | seconds | iter | seconds | iter (ratio) | seconds (ratio) |
| 5000 | 1250 | 0.10 | 0.100 | 59 | 170.54 | 59 | 178.09 | 1.0 | 1.04 |
| 5000 | 1250 | 0.09 | 0.190 | 60 | 190.78 | 23 | 81.75 | 0.4 | 0.43 |
| 5000 | 1250 | 0.08 | 0.379 | 60 | 209.50 | 23 | 88.74 | 0.4 | 0.42 |
| 5000 | 1250 | 0.08 | 0.630 | 61 | 235.14 | 23 | 97.53 | 0.4 | 0.41 |
| 5000 | 1250 | 0.07 | 0.888 | 61 | 297.15 | 22 | 111.51 | 0.4 | 0.38 |
| 5000 | 1250 | 0.07 | 1.112 | 61 | 304.91 | 22 | 120.28 | 0.4 | 0.39 |
| 5000 | 1250 | 0.07 | 1.289 | 61 | 320.62 | 21 | 110.85 | 0.3 | 0.35 |

| p | n | $\lambda$ | NZ% | Cold | | Warm | | Warm/Cold | |
|---|---|---|---|---|---|---|---|---|---|
| | | | | iter | seconds | iter | seconds | iter (ratio) | seconds (ratio) |
| 5000 | 1250 | 0.07 | 1.421 | 61 | 302.57 | 19 | 100.33 | 0.3 | 0.33 |
| 5000 | 3750 | 0.08 | 0.102 | 60 | 213.08 | 25 | 100.04 | 0.4 | 0.47 |
| 5000 | 3750 | 0.06 | 0.118 | 60 | 213.48 | 22 | 82.87 | 0.4 | 0.39 |
| 5000 | 3750 | 0.05 | 0.173 | 61 | 222.05 | 21 | 82.29 | 0.3 | 0.37 |
| 5000 | 3750 | 0.05 | 0.323 | 61 | 272.23 | 20 | 78.67 | 0.3 | 0.29 |
| 5000 | 3750 | 0.04 | 0.568 | 61 | 245.31 | 20 | 93.27 | 0.3 | 0.38 |
| 5000 | 3750 | 0.04 | 0.856 | 61 | 272.03 | 19 | 94.39 | 0.3 | 0.35 |
| 5000 | 3750 | 0.04 | 1.130 | 62 | 332.37 | 18 | 111.98 | 0.3 | 0.34 |
| 5000 | 3750 | 0.04 | 1.359 | 62 | 312.75 | 17 | 117.58 | 0.3 | 0.38 |
| 5000 | 6250 | 0.08 | 0.101 | 60 | 201.55 | 27 | 113.80 | 0.5 | 0.56 |
| 5000 | 6250 | 0.06 | 0.117 | 61 | 197.77 | 23 | 102.70 | 0.4 | 0.52 |
| 5000 | 6250 | 0.05 | 0.144 | 62 | 188.22 | 22 | 107.47 | 0.4 | 0.57 |
| 5000 | 6250 | 0.04 | 0.308 | 62 | 224.07 | 21 | 97.97 | 0.3 | 0.44 |
| 5000 | 6250 | 0.03 | 0.844 | 62 | 301.96 | 21 | 118.37 | 0.3 | 0.39 |
| 5000 | 6250 | 0.03 | 1.757 | 63 | 338.19 | 22 | 147.29 | 0.3 | 0.44 |
| 5000 | 6250 | 0.03 | 2.828 | 63 | 445.84 | 22 | 171.05 | 0.3 | 0.38 |
| 5000 | 6250 | 0.03 | 3.845 | 63 | 508.81 | 22 | 202.41 | 0.3 | 0.40 |

| ccfista_1 | | | | | | | | | |
|---|---|---|---|---|---|---|---|---|---|
| p | n | $\lambda$ | NZ% | Cold | | Warm | | Warm/Cold | |
| | | | | iter | seconds | iter | seconds | iter (ratio) | seconds (ratio) |
| 5000 | 1250 | 0.10 | 0.100 | 34 | 116.29 | 34 | 122.71 | 1.0 | 1.06 |
| 5000 | 1250 | 0.09 | 0.190 | 34 | 146.35 | 18 | 82.83 | 0.5 | 0.57 |
| 5000 | 1250 | 0.08 | 0.379 | 35 | 185.02 | 18 | 95.04 | 0.5 | 0.51 |
| 5000 | 1250 | 0.08 | 0.630 | 35 | 188.99 | 18 | 111.21 | 0.5 | 0.59 |
| 5000 | 1250 | 0.07 | 0.888 | 35 | 241.79 | 18 | 121.71 | 0.5 | 0.50 |
| 5000 | 1250 | 0.07 | 1.112 | 35 | 261.39 | 18 | 144.55 | 0.5 | 0.55 |
| 5000 | 1250 | 0.07 | 1.289 | 36 | 259.91 | 13 | 96.61 | 0.4 | 0.37 |
| 5000 | 1250 | 0.07 | 1.421 | 36 | 268.20 | 13 | 103.76 | 0.4 | 0.39 |
| 5000 | 3750 | 0.08 | 0.102 | 34 | 131.40 | 23 | 101.62 | 0.7 | 0.77 |
| 5000 | 3750 | 0.06 | 0.118 | 35 | 142.58 | 18 | 83.89 | 0.5 | 0.59 |
| 5000 | 3750 | 0.05 | 0.173 | 35 | 146.11 | 12 | 64.77 | 0.3 | 0.44 |
| 5000 | 3750 | 0.05 | 0.323 | 35 | 204.83 | 12 | 66.64 | 0.3 | 0.33 |
| 5000 | 3750 | 0.04 | 0.568 | 35 | 213.24 | 12 | 78.30 | 0.3 | 0.37 |
| 5000 | 3750 | 0.04 | 0.856 | 35 | 230.24 | 12 | 93.24 | 0.3 | 0.40 |
| 5000 | 3750 | 0.04 | 1.130 | 35 | 232.96 | 12 | 113.18 | 0.3 | 0.49 |
| 5000 | 3750 | 0.04 | 1.359 | 35 | 269.01 | 12 | 124.06 | 0.3 | 0.46 |
| 5000 | 6250 | 0.08 | 0.101 | 35 | 141.06 | 24 | 115.84 | 0.7 | 0.82 |
| 5000 | 6250 | 0.06 | 0.117 | 35 | 138.92 | 18 | 100.40 | 0.5 | 0.72 |
| 5000 | 6250 | 0.05 | 0.144 | 35 | 140.20 | 18 | 107.38 | 0.5 | 0.77 |
| 5000 | 6250 | 0.04 | 0.308 | 35 | 185.94 | 17 | 102.35 | 0.5 | 0.55 |
| 5000 | 6250 | 0.03 | 0.844 | 36 | 252.19 | 18 | 147.24 | 0.5 | 0.58 |
| 5000 | 6250 | 0.03 | 1.758 | 36 | 314.23 | 18 | 196.23 | 0.5 | 0.62 |
| 5000 | 6250 | 0.03 | 2.828 | 36 | 477.69 | 18 | 230.70 | 0.5 | 0.48 |
| 5000 | 6250 | 0.03 | 3.845 | 36 | 551.26 | 18 | 282.35 | 0.5 | 0.51 |

## A.8   Other Methods

### A.8.1   Dual problem of CONCORD

Formulating the dual using the matrix form is challenging since the KKT conditions involving the gradient term $S\Omega + \Omega S$ do not have a closed form solution as in the case of Gaussian problem in [2]. Therefore, we consider a vector form of the CONCORD problem by defining two new variables $x_1 \in \mathbb{R}^p$ and $x_2 \in \mathbb{R}^{p(p-1)/2}$ as

$$x_1 = (\omega_{11}, \omega_{22}, \ldots, \omega_{pp})^T$$
$$x_2 = (\omega_{12}, \omega_{13}, \ldots, \omega_{1p}, \omega_{23}, \ldots, \omega_{2p}, \ldots, \omega_{p-1p})^T. \tag{21}$$

We define two coefficient matrices $A_1, A_2$ as

$$
A_1 = \begin{bmatrix} Y_1 & & & \\ & Y_2 & & \\ & & \ddots & \\ & & & Y_p \end{bmatrix}, A_2 = \begin{bmatrix} Y_2 & Y_3 & \cdots & Y_p & & & & & & & & \\ Y_1 & & & & Y_3 & \cdots & Y_p & & & & & \\ & Y_1 & & & Y_2 & & & & & & & \\ & & \ddots & & & & \ddots & & \ddots & Y_{p-1} & Y_p & \\ & & & Y_1 & & & Y_2 & & & Y_{p-2} & & Y_p \\ & & & & & & & & & & Y_{p-2} & Y_{p-1} \end{bmatrix},
$$

$$(22)$$

where $A_{1\,np\times p}$ and $A_{2\,np\times p(p-1)/2}$ dimensional matrices. Using these definitions, the CONCORD problem (4) can be rewritten as

$$
\underset{x_1,x_2}{\text{minimize}} \; -n\log x_1 + \frac{1}{2}\left\|A_1 x_1 + A_2 x_2\right\|^2 + \lambda\|x_2\|_1. \tag{23}
$$

where, $\log(x_1) = \sum_{i=1}^{i=p} \log(x_{1i})$. We will use $x = \begin{bmatrix} x_1 \\ x_2 \end{bmatrix}$ for simplicity of notation where ever possible.

The transformed CONCORD problem in (23) can be written in composite form using a new variable $z = A_1 x_1 + A_2 x_2$ as

$$
\underset{x_1,x_2,z}{\text{minimize}} \; -n\log x_1 + \frac{1}{2}\|z\|^2 + \lambda\|x_2\|_1
$$
$$
\text{subject to} \; A_1 x_1 + A_2 x_2 = z \tag{24}
$$

The Lagrangian for this problem is given by

$$
\mathcal{L}(x_1, x_2, z, y) = -n\log x_1 + \frac{1}{2}\|z\|^2 + \lambda\|x_2\|_1 + y^T \left(A_1 x_1 + A_2 x_2 - z\right). \tag{25}
$$

Maximizing with respect to the three primal variables yields following optimality conditions (the . notation is adapted from MATLAB to denote element-wise operations),

$$
z - y = 0
$$
$$
-n./x_1 + A_1{}^T y = 0
$$
$$
\lambda\text{sign}(x_2) + A_2{}^T y \ni 0. \tag{26}
$$

Substituting these the dual problem can be written as

$$
\underset{y}{\text{maximize}} \; -n\log n./A_1{}^T y + \frac{1}{2}\|y\|^2 + y^T \left(A_1(n./A_1{}^T y) - y\right)
$$
$$
\text{subject to} \; \|A_2{}^T y\|_\infty \leq \lambda,
$$

or equivalently

$$
\underset{y}{\text{maximize}} \; \frac{1}{2}\|y\|^2 - n\log\left(A_1{}^T y\right) + c
$$
$$
\text{subject to} \; \|A_2{}^T y\|_\infty \leq \lambda, \tag{27}
$$

where, $c = n\log n - n^2$ is a constant. This problem can also be written in composite form as

$$
\underset{y}{\text{maximize}} \; \frac{1}{2}\|y\|^2 - n\log\left(A_1{}^T y\right) + \mathbb{1}_{\|w\|_\infty \leq \lambda}
$$
$$
\text{subject to} \; A_2{}^T y - w = 0. \tag{28}
$$

The gradient and hessian of the smooth function $h(y) = \frac{1}{2}\|y\|^2 - n\log\left(A_1{}^T y\right)$ is given by

$$
\nabla h(y) = y - A_1(n./A_1{}^T y),
$$
$$
\nabla^2 h(y) = I + A_1\texttt{diag}\left(n./(A_1{}^T y)^2\right) A_1{}^T. \tag{29}
$$

Here, the hessian is bounded away from the semi-definite boundary. Hence the function $h$ is strongly convex with parameter 1. Moreover, on lines of Theorem 3.1, we can show that if $y$ is restricted to

a convex level set $\mathcal{C} = \{y | h(y) \leq M\}$ for some constant $M$, then the function $h$ has a Lipschitz continuous gradient. Note that

$$-n \log\left(A_1{}^T y\right) \leq h(y) \leq M$$
$$e^{-\frac{M}{n}} \leq A_1{}^T y. \tag{30}$$

Therefore, the hessian satisfies

$$\nabla^2 h(y) = I + A_1 \texttt{diag}\left(n./(A_1{}^T y)^2\right) A_1{}^T \preceq (1 + n\rho(A_1{}^T A_1)e^{\frac{2M}{n}})I. \tag{31}$$

To conclude, the dual problem provides an alternate method to prove the $\mathcal{O}(\frac{1}{k})$ and $\mathcal{O}(\frac{1}{k^2})$ rates of convergence for CONCORD problem.

### A.8.2 Proximal Newton's Algorithm for CONCORD

Recall that the hessian of the smooth function $h_1$ as given in 6 is

$$\nabla^2 h_1(\Omega) = \sum_{i=1}^{i=p} \omega_{ii}^{-2} \left[e_i e_i{}^T \otimes e_i e_i{}^T\right] + \frac{1}{2}\left(S \otimes I + I \otimes S\right).$$

The subproblem solved for the direction of descent for the second order PNOPT algorithm is given by

$$\Delta\Omega^{(k)} = \arg\min_W \langle G^{(k)}, W \rangle + \frac{1}{2}\sum_{i=1}^{i=p} \omega_{ii}^{-2} \operatorname{tr}\left(W e_i e_i{}^T W e_i e_i{}^T\right) + \operatorname{tr}\left(WSW\right) + \lambda\|\Omega_{\mathrm{X}}{}^{(k)} + W\|_1. \tag{32}$$

Using these, the matrix version of the second order algorithm is given in Algorithm 3. Here, the subproblem for the descent step is as a huge Lasso problem. This can be solved by standard Lasso packages which uses coordinate descent methods.

---

**Algorithm 3** CONCORD - Proximal Newton Optimization Matrix form (CONCORD-PNOPT)

---

Initialize: $\Omega^{(0)} \in \mathbb{S}_+^p, \tau_{(0,0)} = 1, \Delta_{\mathrm{opt}} = 2\epsilon_{\mathrm{opt}}$ and $\Delta_{\mathrm{term}} = 2\epsilon_{\mathrm{term}}$

**while** $\Delta_{\mathrm{subg}} > \epsilon_{\mathrm{subg}}$ **or** $\Delta_{\mathrm{term}} > \epsilon_{\mathrm{term}}$ **do**

*Compute* $\nabla h_1$:
$$G^{(k)} = \Omega_{\mathrm{D}}{}^{-1} + \tfrac{1}{2}\left(S\ \Omega(k)^T + \Omega^{(k)}S\right)$$

*Compute Newton step:*
$$\Delta\Omega^{(k)} = \arg\min_W \langle G^{(k)}, W \rangle + \frac{1}{2}\sum_{i=1}^{i=p} \omega_{ii}^{-2} \operatorname{tr}\left(W e_i e_i{}^T W e_i e_i{}^T\right) + \operatorname{tr}\left(WSW\right) + \lambda\|\Omega_{\mathrm{X}}{}^{(k)} +$$
$$W\|_1$$

*Compute sufficient descent* $\Delta^{(k)}$:
$$\Delta^{(k)} = \langle G^{(k)}, \Delta\Omega^{(k)} \rangle + \lambda\left(\|\Omega_{\mathrm{X}}{}^{(k)} + \Delta\Omega_{\mathrm{X}}{}^{(k)}\|_1 - \|\Omega_{\mathrm{X}}{}^{(k)}\|_1\right)$$

*Compute* $\tau_k$, *such that* $Q_{con}(\Omega^{(k+1)}) \leq Q_{con}(\Omega^{(k)}) + \alpha\tau_k\Delta^{(k)}$.

*Update:* $\Omega^{(k+1)} = \Omega^{(k)} + \tau_k\Delta\Omega^{(k)}$

*Compute convergence criteria:*
$$\Delta_{\mathrm{subg}} = \frac{\|\nabla h(\Omega^{(k)}) + \partial g(\Omega^{(k)})\|}{\|\Omega^{(k)}\|}, \qquad \Delta_{\mathrm{term}} = \frac{\|f(\Omega^{(k+1)}) - f(\Omega^{(k)})\|}{\|f(\Omega^{(k)})\|}$$

**end while**

---