[Reviews · NeurIPS 2014]

Submitted by Assigned_Reviewer_6

This paper provides two algorithms based on the soft-thresholding method for estimating a penalized pseudo-likelihood graphical model. The coordinate-wise method seems a practical improvement of the current CONCORD method. Overall, this is a worthwhile addition to a booming literature on this issue. The method has computational complexity O(sp2), but clearly also depends on the starting value. Recent literature have stressed the importance of “warm starts”. The authors could add this.

Nevertheless, the underlying ideas (coordinate-wise soft-thresholding and penalized pseudo-likelihood graphical models) are not new. The novelty consists in putting these ideas together in this framework. Moreover, although the authors can perhaps not be held fully responsible, I would have liked to see some discussion on

*Performance of the pseudo-likelihood in non-normal data.
The motivation is the non-normality of lots of real data. However, the quadratic form of the pseudo-likelihood does not seem to be particularly robust to outliers as is claimed by the authors. In fact, the only simulation study in the paper uses normal data, which is clearly disappointing.

* Choice of the tuning parameter.
The paper does not give any hints on how to select the tuning parameter.
Summary: The paper improves the computational complexity of estimating a sparse graphical model. The novelty is combining several known ideas, which is perfectly acceptable. More awareness of the limitations of the original model would have been welcome.

Submitted by Assigned_Reviewer_28

This paper proposes the use of proximal gradient methods for convex pseudo-likelihood based partial correlation graph estimation. This main contribution here is to bring in faster convergence in a framework that can deal with non-Gaussian data. The paper is well written and the combination of approaches proposed in the paper are very promising. One caveat is that the convergence performance of the method is not given in terms of increasing discrepancy with the multivariate Gaussian assumption. There are multiple ways in which this could be assessed. One of the most appealing ones, given that the authors tackle the analysis of molecular data, would be to assess the method with increasing levels of variance heterogeneity. In fact, the authors assess the method with real microarray data, whose fluorescence-based technology alleviates the variance heterogeneity problem. The authors should try the method with more recent high-throughput molecular technology data, such as RNA-seq, which is more challenging in this respect.
Summary: Good paper, clearly written.

Submitted by Assigned_Reviewer_43

This paper addresses the use of pseudo-likelihood for solving (sparse) Graphical Model selection problems. The use of log likelihood for Gaussian Graphical Models has been the subject of considerable research, but this is limited to the Gaussian assumption. This has not been the case for pseudo-likelihood –based methods, and non-Gaussian graphical models. Since the Gaussian assumption is limiting, there are potentially large benefits in this line of work.

At a high-level the formulation is based on a recent approach called CONCORD, which optimizes Qcon, a convex pseudo-likelihood objective. CONCORD uses coordinate-wise descent. The paper proposes proximal gradient techniques to minimize Qcon and provides a theoretical analysis. Proximal gradient has become a common useful tool in machine learning. The methods are tested in learning non-Gaussian graphical models.

The paper derives convergence rates for CONCORD-ISTA/-FISTA which is of significance, it also compares CONCORD-ISTA/FISTA and coordinate-wise minimization. Results suggest that CONCORD-ISTA is much faster than coordinate-wise, specially in high-dimensions. Since their computational complexity is similar O(p^2), a primary reason for this is the possibility of parallelizing the computation in the proposed method, while this is not the case for the coordinate-wise method. The derived converge rates are not fully consistent across pretty much all of the results. It would be great to have further discussion on this.

Summary: Overall, this is an interesting contribution to solving l1 penalized non-Gaussian graphical model selection, using pseudo-likelihood. It extends previous methods pseudo-likelihood methods by providing convergence results and formulating a convex problem (based on CONCORD). The speed ups resulting from the parallel implementation are a good contribution.
Author Feedback
Author rebuttal: Reviewer 28

We thank the reviewer for his/her kind words mentioning that the paper is "clearly written" and that it is "very promising". Indeed, as pointed out by the reviewer, the main focus and contribution of the paper is to achieve faster convergence for CONCORD, a method which can deal with non-Gaussian data.

We had first assessed our method only on real microarray data for comparison with prior work and due to page constraints. We now plan to implement the reviewer's insightful suggestion of trying the method also on high-throughput molecular technology data.

Reviewer 43

We thank the reviewer for the encouraging remarks mentioning that the work is a "good contribution", the need to go beyond the Gaussian framework since it is limiting, and that "there are potentially large benefits in this line of work."

The reviewer is correct in stating that the paper optimizes a convex pseudo-likelihood objective using proximal gradient techniques, which can be parallelized, and also provides theoretical convergence results. As the reviewer points out, the results suggest that CONCORD-ISTA is faster than coordinate-wise in high-dimensions. Indeed the proposed CONCORD-ISTA is faster than competing methods in Tables 1 and 2 across various sample size/dimension/sparsity regimes. The reviewer asks why the derived theoretical convergence rates are not fully consistent across all the results - presumably why CONCORD-ISTA is faster than CONCORD-FISTA in the numerical work when CONCORD-FISTA actually has a higher rate of convergence. The reviewer raises a very important question. We have thought about it carefully and have a full response to this: First note that CONCORD-FISTA uses second order information, so it enjoys a higher rate of convergence of O(1\k^2). However though CONCORD-ISTA has a lower rate of convergence of O(1\k), it can also be implemented with a Barzilai-Borwein(BB) step which numerically uses second order or curvature information. Thus both methods have the capability of using second order information.

The apparent discrepancy between the numerical results and the derived convergence rates can however be explained both theoretically and practically.

From a theoretical perspective, in very high dimensions with limited samples, likelihood surfaces are very flat. In such settings, the benefit of exploiting curvature is limited, especially in comparison with the additional computational time required to calculate second-order quantities. In addition, the theoretical convergence rates assume knowledge of the Lipschitz constant L. Also note that though CONCORD-FISTA theoretically has a higher rate of convergence, the constant in front of the geometric term is 4 times higher than the corresponding rate for CONCORD-ISTA and depend on the unknown quantity L.

From a practical perspective, the theoretical insight can be seen in two different ways: (i) CONCORD-ISTA without the BB step is faster than with the BB step, affirming that the curvature information is not helpful, (ii) In Table 1 for high dimensions (p = 3000, 5000) the difference between CONCORD-ISTA and CONCORD-FISTA is accentuated in lower sample sizes, affirming the theoretical insight regarding curvature. In our practical implementation, in the absence of the Lipschitz constant L we use backtracking which adds to the variability in required time/iteration.

To address the reviewer's excellent question, we intend to add a paragraph explaining the above in the revision of the paper.

Reviewer 6

We thank the reviewer for your kind words and for mentioning that the paper combines ideas in a novel way, for recognizing that there is a practical improvement on the current CONCORD method, and that the contribution in the paper is a "worthwhile addition to a booming literature on this issue."

We agree with the reviewer that more material on the limitations on the original model would be useful. We intend to add this to the revision of the paper. The reviewer points out that using warm starts could possibly improve our method even more. This is a very good suggestion and we thank the reviewer for pointing it out. We have in fact used warm starts in the past but did not include it in the paper due to page constraints. We have taken the advice of the reviewer very seriously and have focused our efforts this week on immediately implementing the warm start approach, which was undertaken for both the ISTA and FISTA for all the simulated cases reported in the paper. We calculate the average of the ratios in order to quantify the merits of using warm starts vs. using the Identity matrix as a starting value. We note that the reduction in time taken can be as much as 25%, and is enjoyed by both ISTA and FISTA. We shall add the specifics to the revision of the paper. The reviewer's insights are spot-on and have served to improve our method.

The reviewer makes a valid point that it would be useful to undertake more numerical work on non-normal data. The real life breast cancer dataset in the paper is a non-normal data with outliers. The original CONCORD paper compares Gaussian-based methods to the CONCORD formulation on non-normal data and verifies the usefulness of the CONCORD framework. The efficacy of the proximal gradients methods over the coordinatewise method for the CONCORD formulation is illustrated in this paper. We have made a concerted effort to implement our proximal gradient approach on T-data. The relative performance is consistent with the Gaussian setting. We shall add the specifics to the revision of the paper. We thank the reviewer for this useful comment.

The tuning parameter can be selected either using BIC criteria like in the original CONCORD paper, or cross-validation. We shall add material in the revision of the paper to elaborate on this aspect of the method. Thank you for pointing this out.